# Temporal Landmarks and Nostalgic Consumption: The Role of the Need to Belong

**DOI:** 10.3390/bs14020123

**Published:** 2024-02-08

**Authors:** Sigen Song, Min Tian, Qingji Fan, Yi Zhang

**Affiliations:** 1School of Economics and Management, Shanghai Institute of Technology, Shanghai 201418, China; songsigen@sit.edu.cn (S.S.); 216121149@mail.sit.edu.cn (M.T.); zhangy@sit.edu.cn (Y.Z.); 2Business College, Yangzhou University, Yangzhou 225009, China

**Keywords:** temporal landmarks, nostalgia, need to belong, nostalgic consumption

## Abstract

This study investigates the influence of temporal landmarks on nostalgic consumption through the mediating role of the need to belong. In particular, the study identifies end landmarks as one of the triggers of landmarks, a phenomenon that has not been studied in the existing nostalgic consumption literature. The research is composed of one pilot study and three experiments to test our research hypotheses. The results show that end temporal landmarks trigger feelings of nostalgia, which leads to nostalgic consumption through the need to belong. This study underscores the mediating role of the need to belong, which plays an important role in leading to nostalgic consumption. Building upon theoretical perspectives on the need to belong, our study enriches the research literature by linking extreme consumer emotional statuses, such as social anxiety, to the consumer need to belong, showing that consumer nostalgic consumption can become a coping strategy that counteracts these negative feelings and helps in regaining connection and supporting social relationship networks. Marketers may use the signs of end temporal landmarks to increase consumers’ nostalgia, which, in turn, will enhance consumers’ need to belong and thus lead to the purchasing and consumption of nostalgic products.

## 1. Introduction

The recent Coronavirus pandemic disrupted our daily lives and triggered various psychological disturbances and emotional distresses, such as social anxiety and, in extreme cases, death anxiety. Social anxiety not only causes distress to an individual’s personal life but also harms economies and social stability [1]. Meanwhile, the special events during these extraordinary times significantly changed consumer behavior, providing both challenges and opportunities for marketers to design appropriate marketing or brand strategies [2,3,4]. The existing literature reveals that negative emotions, such as loneliness, anxiety, insecurity, and social exclusion, often elicit nostalgia, which is characterized by a feeling of pleasure and slight sadness when one thinks about things that happened in the past [5].

Negative emotions such as anxiety, loneliness, and boredom are associated with interpersonal relationships [6]. To compensate for the loss of self-control and interpersonal connections, people may engage in nostalgic consumption by reminiscing about the ‘good old days’ [7]. Nostalgia, as defined by the Merriam-Webster Dictionary [8], refers to “a wistful or excessively sentimental yearning for a return to or of some past period or irrecoverable condition”. A fond memory may elicit feelings of loss and longing mixed with love, warmth, and belonging [9]. As such, when we experience nostalgia, we often feel a deep sense of connection to our past selves and the people and events that shaped our lives [10].

A growing body of research has shown that many different situations and settings can trigger feelings of nostalgia, such as adverse weather [11], social exclusion [12], loneliness, and boredom [13]. While nostalgia is a bittersweet sensation that often brings us both joy and sadness, human memory is a reconstructive process in which nostalgia becomes a warm glow of the past [14]. By actively engaging with our past memories and experiences—for example, listening to ‘oldies but goldies’ songs, reconnecting with old friends, visiting childhood homes or schools or other places with special meanings—we can share stories and memories with loved ones. Accordingly, anxiety can eventually lead to nostalgic consumption which functions to alleviate anxiety and gain some control over a situation that feels out of control. From the social interaction perspective [15], a comparison of the past with the present through indirect interpersonal interactions can lead to nostalgia. For example, the mention of familiar people and events in casual conversations can trigger a desire for fond memories which aid in regaining human connection with others due to the need to belong. Evidence shows that nostalgia for one’s hometown friendships and for intimacy between lovers can all lead to the nostalgic consumption of products that remind us of our previous experiences [16].

Nostalgic consumption has received research attention due to its potential economic and commercial significance, as well as its humanistic value. In particular, nostalgic consumption often involves products and services, including products and brands, thus generating great value in antique collection, cultural creation, museum art, tourism, old songs, and the creation of historical movies and films [17]. Nostalgic consumption is not just about the material value that the product provides, but, more importantly, about the symbolic value or meanings that are associated with products. Consumers who feel a low sense of security often favor nostalgic goods that remind them of the good old times of the past [18]. For instance, research has reported that more consumers gain comfort in revisiting TV programs and music they enjoyed in their youth since such products and services offer a return to beloved characters from the past [14].

Existing research on nostalgic consumption mainly focuses on examining the link between age and the development of consumer tastes over time and the patterns in the nostalgia proneness of consumer preferences [16]. However, few research efforts have been devoted to the situational and temporal events that trigger nostalgia and the mechanism underpinning the impact of nostalgia on consequent consumption, thus calling for an in-depth understanding of such phenomena and gaining insights for marketing practices [19]. To fill such research gaps, this research makes unique contributions by scrutinizing the influences of temporal landmarks on nostalgic feelings and consequent nostalgic consumption. In addition, this study underscores the importance of the need to belong as an underlying mechanism in such a relation. Our rationale for such contributions to the existing literature is outlined below.

Reminiscing about a particular day, e.g., the start, middle or end of the month/year, may elicit different emotions [20,21]. As such, during momentous life experiences (e.g., graduations, holidays, anniversaries), people can be struck by feelings of nostalgia due to such times evoking a certain amount of stress and anxiety. We envision that nostalgic feelings are more likely to occur at some time points rather than others in daily life. When individuals are affected by fear and threats in real life, e.g., death anxiety or the feeling caused by the “end of life”, they will make up for the lack of security by choosing nostalgic items [22]. As such, Stern [23] called the phenomenon of Americans feeling confused and anxious at the end of the 20th century, with its increasing uncertainty, as the “end-of-the-century effect”, which may lead consumers to buy goods that may seem old-fashioned but carry nostalgic feelings. We argue that anxiety increases consumer responses to particular situational cues, such as end temporal landmarks (i.e., end of day, month, and year), which are more closely associated with nostalgic feelings. Such end temporal landmarks signal deadlines or remind us of incomplete tasks, which evokes time pressure and anxiety. Accordingly, this study contributes to the existing literature by understanding the impact of end temporal landmarks on nostalgic feelings and the consequent buying intentions for nostalgic products.

In addition, considering that the consequences of social anxiety can be isolation and loneliness due to the breakdown of interpersonal relationships, including those with family, coworkers, and friends, consumers will have a strong motivation to build connections with others [24,25]. Reminiscing about fond memories not only helps us feel more connected but also bolsters our own sense of self-regard through social bonds. In this way, nostalgia allows us to transport ourselves back to a supportive social context in which we feel connected. Therefore, we further investigate the role of the need to belong as a mechanism underpinning the relationship between end temporal landmarks and consequent nostalgic consumption. The findings from our empirical studies underscore the effect of end temporal landmarks and the mediating role of the need to belong on nostalgic consumption.

The paper is structured as follows. We first develop theoretical conceptualizations and hypotheses based on a comprehensive literature review. Next, we test these hypotheses with three empirical studies. Following a discussion of the results, we report the theoretical contributions, managerial implications, and future research directions.

## 2. Literature Review

### 2.1. Nostalgic Consumption as a Coping Strategy to Counteract Social Anxiety

Anxiety has a paradoxical effect on human behavior. On the one hand, the constraints imposed by the COVID-19 pandemic, such as self-isolation, suppressed immediate materialistic shopping. On the other hand, social anxiety also increases our preference for options that are safe and provide a sense of control [26]. Human emotions often reflect one’s perception of uncertainty and an environment that is out of control, while providing a useful source of information for making decisions. Anxiety may positively influence nostalgic consumption to combat negative feelings of life meaninglessness or loss of control.

When nostalgia occurs, individuals will attempt to alleviate psychological discomfort through certain coping behaviors such as nostalgia consumption. Nostalgia consumption is mainly manifested by consumers’ preference for products, brands, or advertisements with nostalgic elements [27] as a means to counteract our negative emotions and increase feelings of social connectedness. Thus, engaging in nostalgia is an emotional regulation strategy for social exclusion and feelings of meaninglessness [28].

Anxiety is likely to increase nostalgia and eventually increase preferences for nostalgic consumption, since nostalgia can aid in difficult times, help overcome anxiety or fears of mortality, give life more meaning, etc. Nostalgic consumption functions as an emotional compensation mechanism that consumers use to regulate or compensate for negative emotions brought about by complex social situations. Specifically, when consumers feel lonely, they are prone to nostalgic consumption tendencies because nostalgic products connect them to the past and allow them to feel social support, thus counteracting feelings of loneliness. As a result, people are generally guided to consume through the good things of the past, which, in turn, compensates for the negative emotions created by complex circumstances.

Research shows that the tendency towards nostalgia can be a protective factor against depression and anxiety [29]. For instance, nostalgia increases meaning in life, positive affect, self-esteem, perceptions of social support, and optimism [28,30]. When we are anxious, we naturally seek comfort and control over the situation, e.g., using shopping as retail therapy to repair bad moods or alleviate distress. Nostalgia has been found to benefit individuals when they reminisce about details of their own past, often triggered by life changes and milestones, as a way of counteracting loneliness, promoting feelings of belonging, and fostering healthy coping strategies [31]. Reminiscence therapy often uses life histories such as photographs and music to help individuals improve psychological well-being. By viewing albums, listening to old songs or returning to childhood wonders, nostalgia may offer a way to cope during stressful life events.

### 2.2. End Temporal Landmarks and Nostalgic Consumption

Nostalgia, commonly understood as denoting currently missing aspects of one’s lived past [31,32], can be triggered by a wide range of stimuli in daily life, from sensory experiences (music or songs, photographs, tastes or smells, etc.) to significant life events (birthdays, weddings, and family gathering traditions). The impact of nostalgia might depend upon the time perspective governing the emotion, since the same point in time can be framed as either the start or end of a given time period. Construal level theory suggests that temporal distance can effect a change in perspective [33,34]. Thus, activating the start vs. end temporal landmarks would trigger different emotions. For instance, consumer temporal orientation can vary and their reactions may differ when a marketing or advertising stimulus is framed as a start versus an end of a period of time [35,36]. However, it is yet to be known how the variation in time in temporal landmarks affects nostalgia.

Research reveals a “calendar effect” when individuals recall events of the past year, showing that students tend to recall more incidents from the beginning and end of school terms than from other periods [37]. This calendar effect suggests that, when people have to search their episodic memory, they consider their own calendar rhythms. People organize their lives into segments separated by temporal landmarks, referring to those points in time that are more noticeable than others on calendars or personal life timelines [20,36].

Evidence shows that temporal landmarks can significantly influence individuals’ self-image [21] and motivation for self-improvement [20]. For instance, some moments are better beginnings than others as a fresh start effect or morning effect since the beginning of a new period causes people to engage in activities facilitating goal initiation [20]. On the other hand, people tend to perceive end temporal landmarks (e.g., the end of the week, month, semester or year), as with facing a task nearing a deadline, with more urgency. Research shows that college seniors appreciated their college experience more when they were led to see that they had little time (vs. lots of time) left to enjoy it [38]. Such a phenomenon might be explained by the fact that the end of college elicits their fond memories of past years because happiness is more intense when an experience is perceived as unlikely to reoccur [39].

Consumers are subjectively biased in their evaluation of how they feel at the end of an event [40], and an individual’s overall rating of an event is influenced by their mood at that end event [41]. End temporal landmarks reduce the consumer’s positive fantasy and enhance the consumer’s perceived depletion of psychological resources compared to a normal point in time [20]. In particular, end temporal landmarks may raise individuals’ feelings of time pressure and anxiety as they remind them of incomplete tasks nearing their due date. From a social comparison perspective, an individual would feel peer pressure when he or she observes that others are performing better at the end of the temporal landmarks, increasing anxiety. A higher level of anxiety caused by end temporal landmarks may lead to a lower level of self-control, resulting in moodiness [42]. In an extreme case, when one perceives anxiety caused by such end-of-time signals, it is thus likely to trigger nostalgia by reminiscing about the lost ‘good old days’. Individuals at a later life stage, for example, will focus more on the value of living and emotional closeness, eliciting nostalgic feelings [43]. The activation of end-of-day signs accelerates the psychological depletion of consumers’ perceptions [5], thus leading to nostalgia and nostalgic consumption. Accordingly, we develop the following hypothesis:

**H1:** 
*An end temporal landmark, compared to other time points, tends to positively influence nostalgic consumption.*


### 2.3. End Temporal Landmarks and Need to Belong

Humans are social animals with a strong underlying drive for social connection or the need to belong by establishing harmonious interpersonal relationships with others [44]. When individuals perceive anxiety and insecurity, they tend to have a feeling of fear of missing out (FoMO), which creates anxiety about what they might feel if the outcome of an upcoming event could later produce an experience of elation [45,46]. Since FoMO reflects a perceived deficit in psychological needs, such as the need to belong and social relatedness, it would motivate individuals to interact with others to satisfy their social need to belong [47]. Nostalgic consumption allows consumers to experience what other people may enjoy, connecting with other people to avoid the feeling of being left out or left behind.

When facing end temporal landmarks, people engage in self-reflection by comparing their present self with their past self. Dissatisfaction with the past elicits feelings of anxiety and perceived depletion of psychological resources. As a result, they would seek social support or contact with the outside world to maximize their sense of belonging and alleviate anxiety [48]. For instance, the stress and loneliness due to special events or time may evoke our memories of times when life was full of warmth, love, and family togetherness. As such, Chesler [49] coined the term “COVID-19 lockdown nostalgia”, which refers to the phenomenon that people will miss their family togetherness during a scary time. The sense of belongingness helps individuals regulate emotions and overcome stress and anxiety [50,51]. Based on the above reasoning, we contend that end-of-calendar temporal landmarks trigger individuals’ desire for the need to belong. The following hypothesis is developed:

**H2:** 
*End-temporal landmarks have a positive effect on the need to belong.*


### 2.4. Need to Belong and Nostalgic Consumption

The need to belong is a fundamental and extremely pervasive motivation for interactions within an ongoing relational bond [44]. Individuals readily form social attachments and resist the dissolution of existing bonds. Since a lack of attachments is associated with various negative effects on health, adjustment, and well-being, the need to belong has multiple and strong effects on emotional patterns and cognitive processes.

Factors influencing the human need to belong include demographic variables as well as emotional factors such as perceived anxiety and insecurity [52]. Fond memories can help consumers strengthen bonds with others by reminding them of shared experiences and traditions, fostering a sense of belonging, and alleviating feelings of loneliness and isolation. Individuals who feel socially excluded tend to subsequently feel a need to belong and are thus more likely to prefer nostalgic consumption [27]. This is because nostalgic consumption is associated with an affective compensation mechanism when consumers experience feelings of anxiety and loneliness due to the loss of interpersonal relationships. Consumers often fulfill their need to belong through brand communities to receive social support and regain their self-identity [53,54,55]. Nostalgia often revolves around shared rituals and traditions, such as holiday celebrations, vacations, or weekly family dinners. As such, nostalgic products are preferred when consumers feel a need to belong [30], making consumers feel connected via the consumption of related products and services. Some familiar and trusted products or brands are more likely to be consumed due to their traditional and historical heritage connecting them to the past. When using traditional popular brands, consumers feel reconnected with important people from the time associated with these brands [56]. Therefore, the following hypothesis is developed:

**H3:** 
*The need to belong positively influences nostalgia consumption.*


Putting together the aforementioned discussions and hypotheses, end temporal landmarks trigger emotional changes that enhance consumers’ desire for belonging, which, in turn, positively leads to nostalgic consumption. Accordingly, we propose that the need to belong underpins the relationship between end temporal landmarks and nostalgic consumption:

**H4:** 
*The impact of end temporal landmarks on nostalgic consumption is mediated by consumers’ need for belonging.*


## 3. Overview of Studies

In pursuit of our research objectives, which are to investigate the influence of end temporal landmarks on nostalgic consumption through the mediating role of the need to belong, we have meticulously designed and executed four empirical studies, encompassing a pilot study and three distinct experiments.

The pilot study was meticulously crafted to validate the experimental materials that would be employed in the subsequent main studies. It aimed to ensure that the manipulation of nostalgic stimuli was effective and that the measures used to assess nostalgic consumption were reliable. Study 1 operationalized the end temporal landmarks by manipulating the end of the month, a common time marker that signifies the conclusion of a period. This study was designed to empirically test the primary hypothesis (H1), which posits a positive relationship between end temporal landmarks and nostalgic consumption. Building upon the initial findings, Study 2 aimed to enhance the robustness of our research by introducing a different end temporal landmark, specifically the end of the academic semester. This study served as a cross-validation of the effects observed in Study 1, ensuring that the results are not confined to a single temporal context. Study 3 further delved into the nuanced relationship between end temporal landmarks and nostalgic consumption by employing a natural experimental design. Utilizing contrasting images of street scenes—one depicting a nighttime setting and the other a midday scene—the study manipulated the perception of time’s end and mid-point. This approach allowed us to examine the mediating role of the need to belong (H2–4) in the relationship between end temporal landmarks and nostalgic consumption.

All in all, our methodological choices ensure the statistical and methodological relevance of the research findings. Firstly, we employed a randomized experimental design, which helps reduce bias and enhance the generalizability of the results. Secondly, we used various validated measurement tools to assess nostalgic perception, the need to belong, and consumption preferences, all of which have good reliability and validity. Additionally, we applied Structural Equation Modeling (SEM) to analyze the data, a powerful statistical tool that allows for the simultaneous consideration of multiple variables’ relationships, thereby supporting our mediation hypothesis.

### 3.1. Pilot Study

#### 3.1.1. Research Design and Participants

We employed a one-way between-groups design to check the manipulation of experiment materials being used in the three main studies. We recruited 82 participants (Male = 42, average age = 23.83) from a widely used data collection platform in China, the Questionnaire Star. Participants were randomly assigned to the nostalgic and non-nostalgic product conditions.

#### 3.1.2. Procedure

The study materials for nostalgic consumption were developed based on the typical description of nostalgia from previous studies [27]. The experimental materials include following three stimulus products/services:

Restaurant A vs. Restaurant B (to be used in main Study 1). The participants were presented with two fictional restaurants. Restaurant A was described in a nostalgic context: “This restaurant has a historical heritage. You have the opportunity to experience the traditional and unforgettable cuisine of the past 20 years. Every little detail in the restaurant will make you bring back your fond memories of warmth and familiarity”. In contrast, Restaurant B was described in a non-nostalgic context: “This is a newly opened restaurant where you can enjoy a variety of novel dishes. Not only will your taste buds be greatly satisfied, but every little detail reflects contemporary lifestyles”.

MOOD vs. KELL cameras (to be used in main Study 2). Participants were informed that that there were two fictional camera brands, MOOD and KELL. The Mood camera was introduced as a nostalgic brand: “Do you remember our childhood? It was a very special time when were innocent, caring and endearing. Our fond memories of the past are so precious, and the MOOD camera will always be there to capture your most beautiful and memorable moments”. In contrast, the *KELL* Camera was introduced an established brand without nostalgic sentiment: “Live in the moment, experience the beauty scene, and capture the your most beautiful moments. KELL Camera is always with you, recording every happy life event. Live in the moment and don’t have regrets”.

Concert-W vs. Concert-Q (To be used in main Study 3). The participants were asked to imagine that they had received a voucher for a concert ticket, which could only be redeemed for either concert-W or concert-Q. Concert-W was described with a nostalgic theme, “Sounds of Time”, and its advertisement tagline reads, “Relive the songs and tell the story of time passing”. In contrast, the concert-Q was described with a non-nostalgic theme, “Wandering in the Sea of Music”, and its advertisement tagline reads, “Catch the beat notes and weave beautiful melodies”.

#### 3.1.3. Results

A manipulation check of the two conditions (nostalgic vs. non nostalgic) was conducted by asking participants to rate the nostalgic perception of each experiment/control product/service on a seven-point scale, adapted from Loveland et al. [57] (see the Appendix A for the measurement scale).

The results from ANOVA showed that, in all three groups of experiment materials, participants in the experiment (nostalgic) groups, compared to those in the control (non-nostalgic) groups, rated higher for nostalgic perception. Specifically, for Restaurant A vs. Restaurant B (M _Nostalgic_ = 5.27, SD = 0.67, M _Nostalgic_ = 2.88, SD = 0.71; F(1, 80) = 243.76, *p* < 0.01); for MOOD vs. KELL cameras (M _Nostalgic_ = 5.54, SD = 0.60, M _non-nostalgic_ = 2.78, SD = 0.47; F(1, 80) = 536.513, *p* < 0.01); and for concert-W vs. concert-Q (M _Nostalgic_ = 5.32, SD = 0.47, M _non-nostalgic_ = 2.78, SD = 0.61; F(1, 80) = 441.47, *p* < 0.01). Such results demonstrated that the manipulation was successful for each experiment condition and thus deemed to be appropriate for using in the main studies.

### 3.2. Study 1

#### 3.2.1. Research Design and Participants

The objective of Study 1 was to test the effect of the end temporal landmarks on consumer preferences for nostalgic products. A one-way between-group design was conducted which used end temporal landmarks as the independent variable. The dependent measure is the choice of between two restaurants: Restaurant A (nostalgic condition) vs. Restaurant B (non-nostalgic condition), as described in the pilot study.

Because the manipulation used the school calendar for temporal landmarks as a reference frame, our sample frame was college students. We recruited 100 college participants from the Questionnaire Star platform. After removing the 10 invalid responses, the remaining valid 90 participants (male = 38; average age 21) were randomly assigned to the experimental (end temporal landmarks) and control (non-end temporal landmarks) groups, with 45 participants in each group, which is an adequate sample size for experimentation [36].

#### 3.2.2. Procedure

Temporal landmarks were manipulated by the date the study was conducted, which were two consecutive days in 30 November 2022 (end of the month) and 1 December 2022 (beginning of the month), respectively. The manipulation of the temporal landmarks was checked by asking participants to answer the question “Is it the beginning, middle or end of the month?”). The participant all answered correctly and thus the emancipation of temporal landmark perception was successful. Next, participants were asked to compare and make a choice between two stimulus products, MOOD camera vs. KELL camera. Finally, participants filled in their demographic information such as their age and gender.

#### 3.2.3. Results

The results showed that age and gender had no significant differences between the two groups and thus these were removed from further analysis. The results from binary logistic regression analyses reveal that participants in the end temporal landmarks group (30 November), compared to those in the start temporal landmarks group (1 December), had a significantly higher preference for Restaurant-A (nostalgic) than for the Restaurant-B (non-nostalgic) (B = −1.214, SE = 0.446, *p* = 0.007, OR = 0.297). H1 was supported.

### 3.3. Study 2

#### 3.3.1. Research Design and Participants

The objective of Study 2 was to provide a robustness test for the findings in Study 1, with variations of the stimulus product (Camera) and manipulation procedure (activating the perception of temporal landmarks). As in Study 1, a one-way between-group design was used for Study 2. We recruited 92 valid participants from the Questionnaire Star platform (male = 63; average age = 22.6).

#### 3.3.2. Procedure

Differing from Study 1, which used actual dates (start or end of the month), Study 2 was conducted in mid-December, which is also the end of the semester. We first manipulated participants’ perception of temporal landmarks by instructing them to answer the different questions. For participants in the end temporal landmark condition, the participants were asked whether it was the start or the end of the semester. For participants in the non-end temporal landmark condition, the participants were asked to identify whether it is the beginning, middle or end of the month. All participants gave the correct answer and thus the manipulation was successful. The dependent measure was the choice of nostalgic (MOOD camera) or non-nostalgic (KELL camera) product, as described in the pilot study.

#### 3.3.3. Results

The results from the binary logistic regression analysis showed that the preference for the nostalgic product (Mood Camera), compared to the non-nostalgic product (KELL camera) was significantly higher in the end temporal landmarks group (end of semester) than in the non-end temporal landmarks group (mid-month) (B = −2.315, SE = 0.489, *p* = 0.000, OR = 0.099). Age and gender were not different across groups and did not affect the final outcome. Thus, H1 was replicated.

### 3.4. Study 3

#### 3.4.1. Research Design and Participants

The main objective of Study 3 was to examine the mediation effect of the need to belong between the relationship between end temporal landmarks and nostalgic consumption. As with previous studies, Study 3 had a one-way between-subject design, with temporal landmarks as the independent variable and the need to belong as the mediator. We recruited 113 participants from the Questionnaire Star platform, with 100 valid responses (male = 47, average 24.2), randomly assigned to one of two experiment conditions.

#### 3.4.2. Procedure

Differing from the previous two studies, Study 3 used a set of pictures showing the different times of the day to manipulate the perceived temporal landmarks. The eliminate the effects of the natural temporal landmarks, all participants were invited to participate in an experiment on the same day, between 12:00 and 14:00 p.m. Study 3 invited subjects to participate in the experiment from 12:00 to 14:00 noon on the same day.

First, all participants were shown a set of street pictures. Participants in the end temporal landmark condition saw a night scene of the city, while those in the non-end temporal landmark condition saw a midday scene of the city, with all other things in the pictures kept identical. Next, participants were asked to imagine that if they were in the situation shown in the picture, how would they best describe their thoughts at that moment using four words. The results showed that their description of the scene was consistent with the picture they saw. The manipulation was successful. Then participants were instructed to rate their choice preference (on a 7-point scale) between Concert-W (nostalgic consumption) and Concert-Q (non-nostalgic consumption) based on the scenario descriptions in the pilot study. Using 4 as the cutoff point, ratings above 4 points indicated nostalgic consumption and below 4 points indicated non-nostalgic consumption. Finally, participants completed a measure of the need to belong on a 5-point scale adapted from Leary et al. [57]. The Cronbach α coefficient of 0.914 was established, indicating a good reliability for the measure (see the measurement scale in Appendix A).

#### 3.4.3. Results

Binary logistic regression analyses showed that the preference for nostalgia consumption (attending Concert-Q vs. attending Concert-W) was significantly higher in the end temporal landmark condition (viewing the night scene) than in the non- end temporal landmarks condition (viewing the midday scene) (B = −3.599, SE = 0.671, *p* = 0.000, OR = 0.027), while there was no significant differences. H1 was replicated.

A one-way analysis of variance (ANOVA) was further conducted to test the effect of end temporal landmarks on the need to belong, showing that participants in the night scene condition possessed a significantly higher need to belong than those in the midday scene condition (M_end-temporal landmark_ = 3.33, SD = 0.46, M _non-end temporal landmark_ =1.95, SD = 0.51; F(1, 98) = 205.788, *p <* 0.01), and H2 was supported.

In addition, the results of the binary logistic regression analysis showed that the need to belong positively influenced nostalgia consumption (B = 2.369, SE = 0.455, *p* = 0.01, OR = 10.689), supporting H3.

We conducted a mediation test using the bootstrap method [58]. As shown in Figure 1, the end temporal landmarks positively influenced the need to belong (β = 1.388, t(98) = 10.007, *p* < 0.01), which also positively influenced the preference for the nostalgic product (Convert-Q) (β = 0.632, t_(98)_ = 2.826, *p* = 0.006). There was a significant partial mediating effect, showing the indirect effect of the end temporal landmarks on nostalgia consumption via the need to belong, with an effect size of 0.878 (95% CI = 0.063 to 0.475). Thus, H4 was supported.

## 4. Discussion

The present research, composed of one pilot study and three experiments, examined the impact of end temporal landmarks on nostalgic consumption through the mediating role of the need to belong. Studies 1 and 2 provided robust evidence showing the positive influence of end temporal landmarks on nostalgic consumption. Study 3 underpinned the underlying mechanism, revealing that end temporal landmarks had a direct effect on nostalgic consumption (H1) and the need to belong (H2), which, in turn, led to nostalgic consumption (H3). As a result, a partial mediation effect supports the mediating role of the need to belong in such a relationship (H4).

### 4.1. Theoretical Contributions

Consumers often weave nostalgia into their cultural and entertainment consumption, bringing up childhood memories to combat stressful life events or situations and feelings of anxiety and frustration. However, little is known about the antecedents that elicit consumer nostalgia and its underlying mechanism. This research contributes to the existing literature on nostalgic consumption by identifying the situational cues that influence nostalgic consumption. Previous studies mainly focused on demographic variables and individual nostalgia-proneness to nostalgic consumption, but little attention was paid to antecedent factors, such as consumer emotions caused by particular social events in extreme time periods, which may influence nostalgic consumption.

Our empirical findings extend existing temporal landmark research [20,59] by examining the effect of end temporal landmark framing on consumers’ nostalgic consumption. This empirical research advances the understanding of nostalgic consumption, with a focus on the unique phenomena that end temporal landmarks, caused by consumers’ feelings of distress and anxiety, may eventually evoke nostalgia and consequent nostalgic consumption.

Building upon theoretical perspectives on the need to belong and the fear of missing out, our study enriches the research literature by linking extreme consumer emotional statuses, such as social anxiety, to the consumer need to belong, showing that consumer nostalgic consumption can become a coping strategy that counteracts negative feelings and aids in regaining connection and supporting networks in social relationships.

In a nutshell, nostalgia is a sentimental longing and yearning for the past, with a mix of happiness from recalling fond memories and sadness from realizing that those moments are gone forever. Our findings shed light on nostalgic consumption, showing that consumers can gain a connection to the past by purchasing products related to the past, thereby restoring individuals’ sense of belonging. When individuals perceive anxiety and insecurity, they tend to connect with people close to them in order to seek psychological comfort, i.e., they have a stronger need to belong. In such situations, which can be triggered by certain temporal landmarks, people often desire to go back to a certain period in the past through the nostalgic consumption of products with some relevant meaning that gives them warmth and security. Our findings regarding the mediating role of the need to belong underscore the role of consumer relationships that influence nostalgic consumption under extreme circumstances, such as the case of the COVID-19 pandemic.

### 4.2. Managerial Implications

Nostalgia consumption holds significant value in fields such as antique collection, cultural creation, museum art, tourism, old songs, and the creation of historical films. In particular, nostalgic consumption is closely related to the consumption of cultural products and vintage products, where the symbolic values or emotional meanings surpass their utilitarian value [60]. Cultural consumption involves obtaining goods or services with a symbolic value that extends beyond their utilitarian value. Nostalgia marketing strategies may create a sense of nostalgia for products by adopting nostalgic product appearances, using traditional manufacturing technology, and historical retailing brands, thereby stimulating consumers’ nostalgia for good memories of the past. Understanding the effect of end temporal landmarks on nostalgic consumption allows marketers to adjust or choose an appropriate time to launch relevant marketing campaigns for antiques, vintage furniture, old-fashioned clothes, and traditional cuisines. Marketers may use the sign of end-of-calendar temporal landmarks to increase consumers’ nostalgia, which, in turn, enhances consumers’ need to belong and thus leads to the purchasing and consumption of nostalgic products.

Since the effect of end temporal landmarks on nostalgic consumption is often associated with consumers’ feelings of stress or anxiety at various social situational landmarks or life events at different times, marketers can convert challenging situations during such natural or social events into business opportunities by designing appropriate marketing strategies for nostalgic consumption. Understanding that negative emotions may be triggers of nostalgia has important managerial implications in harnessing its potential to enhance human well-being and design appropriate nostalgic marketing strategies. For instance, one of the most effective ways to harness the power of nostalgia for greater happiness is by actively revisiting our past memories. This can involve a wide range of activities, such as looking through old photos albums, scrapbooks or digital archives that remind us of our past experiences and accomplishments. As such, while COVID-19 has caused stress and anxiety for humans, it has also changed consumer lifestyles and consumer journeys [61,62], which has important managerial implications for retailers to market nostalgic products. Accordingly, some familiar and trusted products or brands are more likely to be chosen, driven by consumer anxiety, insecurity, and uncertainty.

Our finding that consumers’ need to belong plays an important role in explaining the nostalgic consumption mechanism also sheds light on strategic development when designing advertising or promotion campaigns. For instance, as one of the emotional appeals, nostalgic appeal using storytelling can effectively arouse consumers’ need to belong, which helps consumers build a connection with the brand, brand community, and other social networks [63,64]. Retailers of nostalgic products/services can design their products and advertising campaigns with nostalgic appeal by reconnecting with people or brands from the past through the purchase of products, bringing back fond memories. For example, Holbrook and Schindler [16] found that music has a positive effect on consumers’ nostalgic consumption. This is because sensory input mechanisms associated with the perception of good things from the past, such as music, smells, and food, through tactile, gustatory, and auditory mechanisms, can, in turn, elicit nostalgia.

In this study, we investigated the impact of temporal landmarks on nostalgic consumption, particularly through the mediating role of the need to belong. Our findings resonate with previous research that emphasizes nostalgia as an emotional regulation strategy, helping individuals cope with social anxiety and uncertainty. For instance, Routledge et al.’s research indicates that nostalgia can serve as a meaning-making resource, assisting individuals in finding significance during life transitions [28]. Furthermore, Wildschut et al.’s study points out that the content, triggers, and functions of nostalgia may vary across cultures, suggesting that nostalgic consumption could be influenced by cultural factors [30]. For example, Chinese research often emphasizes the connection between nostalgic consumption and collective memory and a sense of security [22], which also resonates with the mediation of the need to belong in this study. However, in some western cultures, nostalgic consumption may be more closely associated with personal growth experiences and popular culture [65,66]. This difference suggests that, when designing cross-cultural marketing strategies, it is necessary to consider the cultural background of the target market and the consumers’ different interpretations of time, history, and personal experiences.

## 5. Limitations and Future Research Directions

Given the limited scope of this research, which focuses on the impact of end temporal landmarks on nostalgic consumption, future research may expand the scope by investigating other situational variables as antecedents to nostalgic consumption. This would further validate and enrich our understanding of the nostalgic consumption phenomenon and its underlying mechanisms and boundary conditions.

While experimental designs with variations of stimulus situations are appropriate for our research objective to achieve high internal validity with a better control of extraneous variables, future research may use other research methods, such as field studies, to validate our findings. Similarly, because the experimental scenarios used in our experiment are designed at the end of the semester, our sample mainly includes college-aged students. Future research may expand the sample to cover a variety of age groups to enhance ecological validity, which will increase the managerial implications and practical guidelines for retailers.

## Figures and Tables

**Figure 1 behavsci-14-00123-f001:**
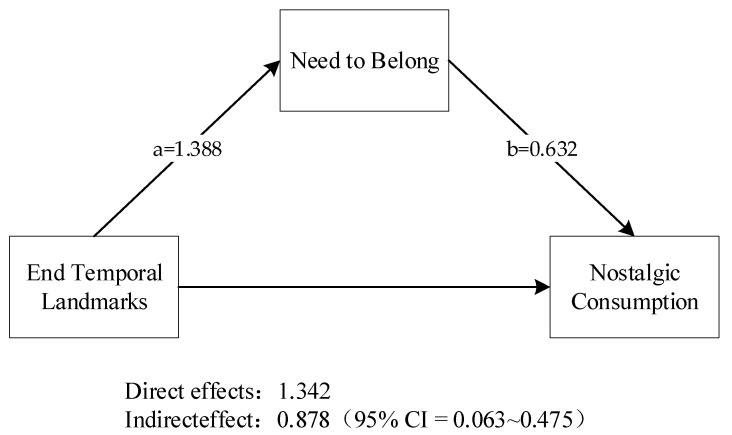
Test of mediating role of need to belong.

## Data Availability

The data presented in this study are available upon request from the authors.

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
