# Peer review of "Temporal Landmarks and Nostalgic Consumption: The Role of the Need to Belong"

_behavsci, 2024, doi:10.3390/bs14020123_

Round 1
Reviewer 1 Report
Comments and Suggestions for Authors
A very interesting and well-presented set of experiments.
Temporal effects are quite interesting. Applications for marketers are significant and could be expanded.
Additional specific comments:
This set of experiments shows the effects of Temporal Landmarks such as the end of the month or the beginning of a college semester on whether subjects prefer nostalgic products or services to more contemporary substitutes. The need for belonging is shown to have a moderating effect on nostalgic consumption preference.
The study of temporal landmarks ie. specific points in time that trigger emotions and product preference is a very interesting psychological construct that as noted by the authors could have significant implications for marketers.
The temporal points selected make the experiments reported of particular interest. While other work has examined points at which one would expect emotional effects such as a holiday this work uses the end – beginning of the month, night – day and the college semester as treatments.
As the authors note in their limitations the subjects are primarily college students. Thus, they could be particularly sensitive to the end of a semester. Students may also be away from home making them more sensitive to nostalgic consumption opportunities and in a social setting making them also sensitive to the need to belong. This is at one reading a study of college students in China. Extending the research to other cultural and national settings is needed in future work.
This is a criticism however that can be leveled against many highly referenced psychological and other academic studies.
Given the caveat above acknowledged by the authors, the conclusions are consistent with the evidence and arguments presented.
References are appropriate.
The tables and figures are well presented in the excepted format.
Comments on the Quality of English Language
A couple of typos easly fixed with quick edit
Author Response
Dear Reviewer:
We are deeply grateful for your thorough review and insightful comments on our manuscript. We have carefully considered your feedback and made the necessary revisions to the original text. Please find our detailed responses to your valuable suggestions in the attachment. We sincerely appreciate your contribution to the improvement of our work.
Best wishes,
Yours sincerely,
All authors

Reviewer 2 Report
Comments and Suggestions for Authors
See PDF file

Author Response

(The authors gave the same response as above.)

Reviewer 3 Report
Comments and Suggestions for Authors
Dear authors, thank you for your research, which proposes an interesting topic of analysis.
However, the study undertaken requires certain additions relevant to the research activity:
- starting from the methodological model chosen for the current presented analyses, to define more clearly the research objectives.
- the objectives should be presented in relation to the hypotheses already proposed so that there is a concrete argumentation of the chosen methodology and of the four empirical studies and three experiments that you have chosen
- please clarify why the chosen methodology can be statistically and methodologically relevant to support the identified results.
Recommendations:
- we suggest that you enrich the introduction with more bibliographic resources and current data as well as adding missing citations or bibliographic references
- we recommend a general revision of the English language for the entire article (for example: in the introduction you have a lot of repetitions)
- in the discussion part, it would be interesting to have a comparative approach with similar studies previously carried out and presented in the literature, perhaps even in different geographical regions, in order to be able to see possible similarities or differences resulting from the analyzed behaviors
Author Response

(The authors gave the same response as above.)

Round 2
Reviewer 3 Report
Comments and Suggestions for Authors
Distinguished authors, thank you for taking into account the recommendations made, I appreciate the useful additions to the research and congratulate you for your work!
Best wishes to all of the authors!